# Analysis of a State Police Academy Menu Cycle for Dietary Quality and Performance Nutrition Adequacy

**DOI:** 10.3390/ijerph191912642

**Published:** 2022-10-03

**Authors:** Bryan Michael Pepito, Jay Dawes, Deana Hildebrand, Jillian Joyce

**Affiliations:** 1Tactical Fitness and Nutrition Lab, College of Education and Human Sciences, Oklahoma State University, Stillwater, OK 74078, USA; 2Department of Nutritional Sciences, College of Education and Human Sciences, Oklahoma State University, Stillwater, OK 74078, USA; 3School of Kinesiology, Applied Health, and Recreation, College of Education and Human Sciences, Oklahoma State University, Stillwater, OK 74078, USA

**Keywords:** nutrition, performance, dietary quality, law enforcement, training academy

## Abstract

Law enforcement officers have high rates of overweight and obesity. With diet as a leading risk factor, training academies present an opportunity for early-career nutrition intervention. Our purpose was to determine the dietary quality (DQ) and performance nutrition adequacy of a state police academy’s cafeteria menu. This cross-sectional content analysis included six weeks (three daily meals, Monday–Friday) of a police academy menu. Nutrient content was determined by portioning menus, gathering food specifications, and performing nutrient analysis. DQ was assessed using the Healthy Eating Index (HEI) 2015. Statistical analyses included independent *t*-tests and Cohen’s d. The total HEI score was 54/100. Subcomponent scores indicating adequacy included added sugar (5/5), total protein (4.97/5) and whole fruits (4.77/5). Seafood/plant proteins (0.33/5), fatty acid ratio (1.31/5), and dairy scores (1.59/10) needed significant improvement. The menu met the recommended intake for 13 of 19 nutrients investigated. Nutrients that did not meet adequacy were calories (% mean difference, needs-menu = 36.7%), carbohydrates (52.3%), vitamins D (82.5%) and E (66.7%), magnesium (44.1%), and potassium (41.8%). The academy menu leaves room for improvement in DQ and shortfall nutrients. By increasing low scores, the overall DQ of the menu will increase and supplement missing nutrients.

## 1. Introduction

Law enforcement officers are responsible for enforcing the law and maintaining public order via investigation and prevention of criminal activities. However, few measures have been taken to protect the health of these public servants. Research suggests that officers have a similar prevalence of overweight and obesity compared to the general population [1,2]. In the United States (US) alone, body mass index (BMI) assessments have placed approximately 38.7% of law enforcement officers in the obese (BMI > 30) weight category [3]. BMI is subject to measurement error and has also been shown to underestimate weight status (i.e., categorize an individual in a healthier or lower BMI group than they would be via other more direct assessments of body fat) in values greater than 28 [4]. Thus, the previously mentioned assessments could potentially be underreporting the amount of law enforcement officers suffering from obesity.

Obesity has been known to be linked to various chronic diseases, such as pulmonary diseases [5], cardiovascular disease (CVD) [6], diabetes mellitus type two [7], and different forms of cancer in the general population [8]. Existing research with law enforcement officers shows that this population is just as susceptible to the aforementioned chronic diseases as the general population [1,9,10,11], which may not be surprising considering the similarity in obesity prevalence previously stated to that of the general population. To address obesity pandemic that is currently plaguing the nation, one must examine two of the most important contributors to weight status and dietary quality.

Dietary quality (DQ) refers to a way of describing and gauging an individual’s diet in terms of health. By analyzing an individual or population’s DQ, researchers are able to better evaluate diet and dietary habits, analyze efficacy of nutrition interventions, and assess risk of chronic diseases [12]. The various factors that can influence DQ can be divided into demographic related factors (geographic location [13], socioeconomic status [14], ethnic background [15], age [16], etc.) and non-demographic related factors (non-homeostatic variables [17], communal eating [18], social media [19], perception towards nutrition [20], etc.). For measuring DQ, numerous indices have been developed based on adherence to scientific and/or government guidelines [21] Scoring for the indices will typically assess moderation, balance, variety, and adequacy of specific food groups and nutrients [22]. Out of all the indices, the Healthy Eating Index (HEI) is consistently updated, as it assesses DQ in relation to the Dietary Guidelines for Americans which updates every 5 years, and has been shown to be both valid and reliable amongst the US population for people two years and older [23]. Positive DQ scores indicate good adherence to government or scientific guidelines for a healthy diet; good adherence has been shown to have favorable effects on weight status, while poor adherence will yield the opposite [24].

With diet as a leading risk factor for obesity, early career nutrition interventions could be beneficial for establishing long-term healthy eating habits and combating the high rates of overweight/obesity found in LEOs. Training academies present as an early career intervention opportunity. By targeting LEOs early in their training phase, relevant staff members may be able to induce beneficial long-lasting dietary habits. While training academies should focus on DQ for health purposes, it is also important to focus on performance nutrition needs and adequacy of meeting those needs to ensure optimal physical performance during training and recovery after. Thus, our purpose was to determine the DQ and performance nutrition adequacy of a state police academy’s cafeteria menu. Results of performing this type of menu analysis could provide invaluable and actionable information to training academy leaders to improve performance and health during the academy and beyond.

## 2. Materials and Methods

The study was a cross-sectional content analysis that looked at the performance nutritional adequacy and DQ of a 6-week sample menu from a state police training academy. The menu included 30 days and consists of breakfast, lunch, and dinner provided to academy cadets Monday through Thursday; on Friday, only breakfast and lunch was provided. Saturday and Sunday were not included because the academy does not provide meals on weekends. The analyzed menus were provided by the academy’s food service manager for July and August of 2020 and is an example of what cadets are fed at the academy for the first six weeks. During these first six weeks, police cadets have their meals pre-portioned for them by cafeteria staff members trained in serving specific portion sizes.

The menu provided to the researchers included a list of foods served without specified serving sizes and lacking some necessary details for accurate nutrient analysis (e.g., the flavor and fat content of milk). Thus, researchers acquired a detailed purchase order from the food service manager to assist with food item details. Researchers also worked with the food service manager to determine serving sizes such that they accurately reflected what cadets were served. While preparing the menu for analysis, all assumptions concerning specific food ingredients and self-service items were reviewed by the food service manager to ensure accuracy. Approval from Oklahoma State University’s Institutional Review Board was not required because this study does not involve human subjects.

### 2.1. Dietary Quality Assessment

Once nutrient content was analyzed, DQ, or the healthfulness of the menus, was assessed using the Healthy Eating Index (HEI) 2015 [25]. The HEI-2015 scores overall DQ on a 100-point scale, including 13 food group and nutrient scoring components as well as general healthy nutrition concepts of adequacy, moderation, variety, and balance. The HEI-2015 has been deemed a reliable and valid measure of DQ for anyone two years of age or older and to whom the Dietary Guidelines for Americans apply to [23]. HEI scores were calculated through a Microsoft Excel spreadsheet that was developed by one of the investigators [26].

### 2.2. Performance Nutritional Adequacy

Individual menu days including breakfast, lunch, and dinner were entered into nutrient analysis software (ESHA Research, Food Processor, version 11.6.441, 2018, Salem, OR, USA) to determine nutrient content. The menu’s nutrient content was compared to the performance nutrition needs of the average academy cadet to determine adequacy of the menu in supporting daily physical training performance. Macronutrient calculations and micronutrient were derived from the Academy of Nutrition and Dietetics (AND) [27]. The AND also determined which micronutrients were relevant for performance needs and stated that micronutrient adequacy should be based off of the DGA’s recommendations for the general populace. The vitamins examined were vitamins A, C, D, E, B1, B3, B6, and B12. The minerals examined were calcium, magnesium, phosphorus, potassium, selenium, sodium, and iron.

To determine performance nutrition needs of the average cadet, annual cadet anthropometrics were provided by the academy’s wellness and fitness coordinator. These data consisted of the average cadet’s age, weight, height, and BMI for the years of 2016–2019. Recommendations are based off the needs of the average police cadet in 2018 because the other years’ average cadet had BMIs indicating near obesity. Using a higher BMI population would have changed the purpose to focusing on weight loss rather than general assessment for adequacy for performance, so we used the BMI that was borderline normal. The average cadet was 30.6 years old, 177 cm, 83.3 kg, and had a BMI of 26.5. The coordinator also provided information on typical physical training regimens to guide determination of macronutrient and caloric needs.

### 2.3. Statistical Analyses

Descriptive statistics included mean, standard deviation, percentages for proportions, and 95% confidence intervals for all nutrients, HEI total score, and HEI component subscores. After ensuring parametric test assumptions were met, independent *t*-test was used to determine performance nutrition adequacy of the academy menu via differences in nutrient content between the menu (overall 30-day mean) the performance macro- and micronutrient needs of the average cadet (as specified above). Cohen’s d was used to determine effect size of significant differences. Level of significance was set at *p* < 0.05. All statistical analyses were completed using SPSS statistical software (version 25, standards, IBM, Armonk, NY, USA).

## 3. Results

### 3.1. HEI-2015 Scores of the Academy Menus

Table 1 shows descriptive statistics for DQ of the academy menu for thirty days. The mean daily total HEI score was 53.9, which indicates that the overall diet quality provided by the menu is in need of improvement (good: >80, needs improvement: 50–80, poor: <50) [28]. In almost all categories, the range and standard deviation indicate that scores greatly varied on a daily basis. The HEI scoring components for which this large variation was not observed were whole fruit, total protein, and added sugar, indicating that these food groups are more consistently provided and that this nutrient is consistently not provided.

Figure 1 visually presents the overall HEI score for the menu, as well as the subcomponent scores. Based on percentage of max score achieved, areas in which the menu is scoring high include added sugar (100%), total protein (99.5%), and whole fruits (95.4%), followed closely by total vegetables (75.4%) and dark greens and legumes (70.1%). Areas in greater need of improvement include seafood/plant proteins (6.7%), fatty acid ratio (13.1%), dairy (15.9%), saturated fat (43.7%) and sodium (46.7%). The remaining three categories of total fruit (63.8%), refined grains (53.3%) and whole grains (61.0%) also leave room for improvement.

### 3.2. Performance Nutrition Adequacy of the Academy Menu

Table 2 describes the nutrient content of the academy menu and compares it to the performance needs of the average cadet [27]. The findings indicate the menu is adequate in provision of protein, fat, vitamin A, B vitamins, vitamin C, calcium, iron, phosphorus, selenium, and magnesium (*p* > 0.05). The findings also indicate the menu alone is not meeting average cadet needs for calories (mean difference = 36.7%, *p* < 0.015, partial eta squared = 0.19), carbohydrates (mean difference = 52.3%, *p* < 0.00, partial eta squared = 0.36), vitamin D (mean difference = 82.5%, *p* < 0.00, partial eta squared = 0.98), vitamin E (mean difference = 66.7%, *p* < 0.00, partial eta squared = 0.52), magnesium (mean difference = 44.1, *p* < 0.02, partial eta squared = 0.16), and potassium (mean difference = 41.8, *p* < 0.04, partial eta squared = 0.14). Despite the small effect size for differences in calories, magnesium, and potassium, the percent differences range from 36.7 to 44.1%, which is a clinically large difference. The difference per effect size was moderate for carbohydrates; however, this too was clinically significant with a percent difference of 52.3%. Finally, the effect sizes indicate a large difference in vitamin D and vitamin E between cadets needs and menu provision, which is supported by clinically large percent differences in provision.

## 4. Discussion

The purpose of this study was to determine the DQ of a state police academy’s cafeteria menu and if the menu provided adequate nutrition for performance needs. The overall DQ of the menu, with a score of 54, is in need of improvement. The subcomponent scores indicate cadets are receiving adequate daily protein and whole fruits and low amounts of added sugars, while the overall menu lacks sources of seafood and plant proteins, unsaturated fats, and dairy products. Other subcomponents, related to vegetable, grain, and total fruit intake, could also be improved. Additionally, the menu met the recommended intake for thirteen nutrients out of the nineteen nutrients investigated. However, the six nutrients that did not meet adequacy are integral for performance needs; these six nutrients were calories, carbohydrates, vitamins D and E, magnesium, and potassium [27].

For DQ, the subcategories that were in most need of improvement were seafood and plant proteins, unsaturated fats, and dairy. The lack of seafood and dairy may correlate with the large vitamin D deficiency. The ratio of poly- and monounsaturated fats to saturated fats indicates that there is a larger amount of saturated fats in the diet; this may lead to long-term cardiovascular diseases and other chronic comorbidities if this becomes a long-term dietary habit [6]. However, feasible solutions are available to increase these scores. The substitution of seafood, legumes, or tofu for land animal proteins throughout the week would greatly increase the points from seafood/plant proteins. Reducing cheese consumption and substituting oils for solid fats would also increase the points from fatty acids. For dairy, adding 8 oz of milk every morning would also improve scores.

Furthermore, the HEI is meant to be reflective of the DGA; by having a high DQ score on the HEI, one would have a diet that consists of a better food group and nutrient composition. Thus, by improving the DQ score of the academy menu, we can assume that deficient micronutrients would improve. For instance, substituting seafood for some protein options in meals and increasing dairy provision would increase vitamin D intake. Carbohydrates, vitamin E, magnesium, and potassium would also increase with the addition of more vegetables and seafood and plant proteins. In order to increase calorie and carbohydrate intake, snacks should be, and in actuality are, provided between meals. It should be noted that snacks were not included in this assessment due to lack of information and significant variation between cadets.

Looking deeper into the implications of the menu’s nutrient content, a lack of calories and carbohydrates indicates that the cadets who are not receiving additional sources of nutrition (i.e., snacks provided by the program) may not be achieving peak performance due to undernutrition. The vitamins and minerals that were lacking are integral for performance in an athletic population. Vitamin D deficiencies in athletic populations have been associated with increased risk for morbidities, such as osteoporosis and osteomalacia [29,30]. Vitamin E serves as an antioxidant, which has been suggested to have beneficial effects on athletic populations by reducing muscle damage, fatigue, and improving performance [31]. Magnesium is a mineral that helps in maintaining normal muscle function, heart rhythm, blood glucose levels, and blood pressure; all these traits are commonly studied to help aid athletic performance [32]. Lastly, potassium acts in conjunction with sodium in the Na+/K+-pump; this pump is integral for active transport of glucose or amino acids and regulating cellular volume. Additionally, it aids in skeletal, smooth, and cardiac muscle contractions [27].

In relation to areas for future research and practice, researchers could determine the number of snacks provided to cadets on average and include them in nutrient and DQ analyses. This process would require a detailed list of snack brands, portion sizes, and average times that the snacks were provided to each cadet. Additionally, this type of menu analysis could be performed with other academies to determine if these results are consistent across US LE academies. Finally, for practice, public health practitioners, particularly Registered Dietitians, could create suggested nutrition standards for menus, much like the National School Lunch Program. Practitioners could also offer this type of menu analysis as a service to help academies improve the healthfulness of their menus.

A major strength of this study comes from the reporting of items listed on the menu. The dining facility manager provided product purchasing orders and recipes, which allowed the researchers to identify specific ingredients and brands used at the facility. Additionally, the dining facility manager proofread the detailed, portioned menu created by the researchers to ensure that portion sizes and food specifications were truly reflective of what cadets would be served. Additionally, the HEI was used to measure DQ, which has been shown to be both valid and reliable [23]. The methodology used for menu analysis has been utilized in previous studies [26,33]. and undergone peer-review screening.

There are several limitations to be noted in this assessment. A major limitation was the cafeteria menu was not necessarily indicative of what LE cadets truly consumed. Rather than measuring what was consumed, it measured what was served. Thus, if a cadet was not consuming enough food, the analysis was unable to account for it. Additionally, cadets consumed three pre-portioned meals on Monday through Thursday and for both breakfast and lunch on Friday. However, the menu did not account for meals on Saturday, Sunday, and anything consumed after lunch on Friday. Assuming the cadet eats three meals on Saturday and Sunday and another on Friday, the analysis lacked 1/3 of a cadet’s meals for the week. Furthermore, the menu did not account for any additional snacks provided by the academy. As with all DQ indices, the HEI did not account for high limits for adequacy subcategories. For example, an individual that consumes 16oz of protein for every 1000 kcal would receive full points in the protein subcategory, despite this being an extreme amount of protein.

An additional limitation is that there was a salad bar available for lunch and dinner, which was not included in the analysis. Thus, both vegetable subcategories and both fatty acid subcategories may not be truly reflective of the true value. The salad bar was not included in the analysis because there was a lack of data concerning portion sizes and types of vegetables consumed. Condiments for foods were also not included in the analysis for similar reasons. Subcategories that may be affected by this exclusion are added sugars, sodium, MUFA: PUFA ratio, and saturated fats.

## 5. Conclusions

Based on the score provided by the HEI, the menu provided by this specific academy leaves room for various improvements. While it does provide adequate nutrition for a majority of the performance-related nutrients, it lacks in several key nutrients that are necessary for peak performance. By focusing on increasing HEI subcomponent scores, overall DQ of the menu will increase and could potentially make up for the nutrients that are inadequate. Considering the high rates of obesity among LE populations and diet as a leading risk factor, academy menus present as an early-career intervention opportunity through which to have a potential lasting impact on cadet health and performance. Providing feedback similar to that of this analysis could be invaluable.

## Figures and Tables

**Figure 1 ijerph-19-12642-f001:**
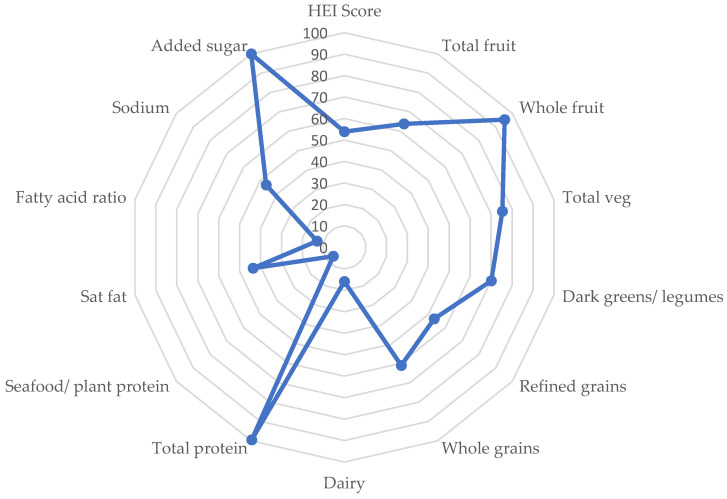
Percentage of max score achieved by the menu for the overall HEI score and all subcomponent scores (%).

**Table 1 ijerph-19-12642-t001:** Descriptive statistics for DQ of the academy menu.

HEI-2015 Scoring Components (Max Score Possible)	Mean	Standard Deviation	Percent of Max Score Possible	Range
Total fruit (5)	3.19	1.05	63.8%	1.65–5.00
Whole fruit (5)	4.77	0.45	95.4%	3.30–5.00
Total vegetable (5)	3.77	1.23	75.4%	1.09–5.00
Dark green/legumes (5)	3.51	2.03	70.1%	0.00–5.00
Whole grains (10)	6.10	4.82	61.0%	0.00–10.00
Dairy (10)	1.59	2.05	15.9%	0.00–8.09
Total protein (5)	4.97	0.14	99.5%	4.21–5.00
Seafood/plant protein (5)	0.33	1.27	6.7%	0.00–5.00
Fatty acid ratio (10) [(MUFA + PUFA)/SFA]	1.31	1.52	13.1%	0.00–4.90
Refined grain (10)	5.33	4.50	53.3%	0.00–10.00
Sodium (10)	4.67	3.21	46.7%	0.00–10.00
Added sugar (10)	10.00	0.00	100%	10.00–10.00
Saturated fat (10)	4.37	3.03	43.7%	0.00–10.00
Total score (100)	53.9	9.55	53.9%	35.04–68.3

**Table 2 ijerph-19-12642-t002:** Comparison of nutrient content between the academy menu and performance needs of the average cadet.

Nutrient	Average Cadet Performance Needs (Mean)	Average Menu Yield (Mean ± sd)	Mean Difference (Needs-Menu)	% Difference (Mean Diff /Needs × 100)	*p*-Value	Effect Size (Partial eta Squared) ^+^
Calories (kcal)	2900	1834.98 ± 405.38	1065.02	36.7	0.015 *	0.19
Carbohydrate (g)	415	197.95 ± 52.80	217.05	52.3	<0.001 *	0.36
Protein (g)	130	101.05 ± 21.51	28.95	22.3	0.20	0.06
Fat (g)	80	72.73 ± 20.27	7.27	9.1	0.723	0.004
Vitamin A (mcg)	900	723.36 ± 450.28	176.64	19.6	0.70	0.005
Vitamin B1 (mg)	1.2	1.25 ± 0.41	−0.05	−4.2	0.91	0.00
Vitamin B3 (mg)	16	21.80 ± 9.43	−5.80	−36.3	0.55	0.012
Vitamin B6 (mg)	1.3	2.16 ± 0.89	−0.86	−66.2	0.35	0.03
Vitamin B12 (mcg)	2.4	3.13 ± 1.16	−0.73	−30.4	0.54	0.013
Vitamin C (mg)	90	99.77 ± 50.72	−9.77	−10.9	0.85	0.001
Vitamin D (mcg)	15	2.62 ± 0.32	12.38	82.5	<0.001 *	0.98
Vitamin E (mg)	15	5.00 ± 1.76	10.00	66.7	<0.001 *	0.518
Calcium (mg)	1000	590.66 ± 215.58	409.34	40.9	0.07	0.107
Iron (mg)	8	11.94 ± 2.85	−3.94	−49.3	0.19	0.06
Magnesium (mg)	400	223.52 ± 74.64	176.48	44.1	0.02 *	0.157
Phosphorus (mg)	700	1332.98 ± 432.25	−632.98	−90.4	0.16	0.067
Potassium (mg)	4700	2733.64 ± 890.74	1966.36	41.8	0.04 *	0.14
Selenium (mcg)	55	112.48 ± 40.05	−57.48	−104.5	0.17	0.064
Sodium (mg)	1500	3181.74 ± 1230.6	−1681.74	−112.1	0.19	0.059

* *p*-values < 0.05 were considered significant. ^+^ effect size values <0.2 were considered small, 0.2–0.5 were considered medium, and >0.5 were considered large.

## Data Availability

Data are proprietary to the group with which we worked, and thus cannot be shared.

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
