# Peer review of "Analysis of a State Police Academy Menu Cycle for Dietary Quality and Performance Nutrition Adequacy"

_ijerph, 2022, doi:10.3390/ijerph191912642_

Round 1

Reviewer 1 Report

The manuscript is clear and relevant for the field. It is scientifically sound and the experimental design is appropriate to test the hypothesis. They properly show the data and they are easy to interpret and understand. The conclusions are consistent with the evidence and arguments presented but I think don’t provide sufficient background and must be include more relevant references. The question is original and well-defined and the results could provide an advancement of the current knowledge. The paper can be accepted after minor revisions in particular in the introduction and conclusions.

Author Response

Thank you for your time in reviewing this manuscript and for your thoughtful comments. Unfortunately, there is minimal research in the area of law enforcement nutrition habits. Thus, we have included as much relevant research in the introduction as possible. Please let us know if a different addition was in mind.

Reviewer 2 Report

Introduction : 

Authors should clearly state the goals of this study at the end of the introduction.  

Results : 

3.1. HEI-2015 Scores . Please correct the typo in the part.

 table, figure

The numerical representation of the data should be the same.

Discussion

(1) compare gathered data with the results by other authors

(2) formulate implications of the results of their study and studies by other authors

References :

Kindly follow the journal format for the references.

Author Response

Thank you for your time in reviewing this manuscript and for your thoughtful comments. Please see how comments were addressed below:

A statement has been added following the purpose statement at the end of the introduction to state the broader goal of this study.

Table 1 has been updated to include numerical values from figure 1.

In terms of the discussion, there unfortunately is very little data on law enforcement officer nutrition habits in the US.  There are no known studies on dietary quality to date to which to compare.

Reference style has been fixed using the EndNote link on the IJERPH author guideline website.

Reviewer 3 Report

This reader recommends adding more clarity to the stated criticism of BMI. The bolded section below could be restated for clarity.

lines 37-38 BMI is subject to measurement error and has also been shown to underestimate weight status in values greater than 28

It is not clear to me that this statement is correct - does not the condition of obesity raise the risk of NCDs?

Line 43-44 - The research shows that this population is just as susceptible to the aforementioned chronic diseases as the general population [1,9-

Line 46- two of the most important contributors to weight status, nutrition status, and diet quality

More frequently the two most important contributors are level of active lifestyle and dietary patterns. - nutrition status and dietary quality are most often seen as subsets of the same contributing factor

Line 52-53- non-demographic related factors (non-homeostatic variables [17], does this include food environments? They have been proven to greatly impact food availability and food consumed.

Like the authors noted above about the criticism of the use of BMI, many critiques can be found of the Dietary Guidelines for Americans

Line 59-60 Dietary Guidelines for Americans which updates every 58 5 years and has been shown to be both valid and reliable amongst the US population for people two years and older [23].

The rationale for using the two terms: nutrition status, and diet quality are not illuminated for the reader in the final two paragraphs of the introduction. If sufficiently distinct in this study, readers may benefit from understanding this distinction and application in this study.

This reviewer appreciates the complication but wonders about the methodology and the researchers not seeing or assessing examples of servings that were plated and the amount of food left unconsumed (or wasted).

Line 113 this data - should this be these data. Are data not plural?

Line 114-115 unclear - what need of cadets? And for normal cadet features does gender factor into weight and height and BMI?

Line 155 - what is used to determine whether meeting average cadet needs for calories? For the food to not meet caloric levels on a daily basis seems surprising as the contrary would have been more likely. 2900 calories may be excessive for sedentary activity like attending courses for weeks. But it may be a good proxy for a more active lifestyle or an athletic population

It may be helpful to note the importance of the quantity and the quality of the six nutrients that did not meet adequacy are integral for performance needs,; 179 these six nutrients were calories, carbohydrates, vitamins D and E, magnesium, and po- 180 tassium [27]

Author Response

Thank you for your time in reviewing this manuscript and for your thoughtful comments. Please see the stars (*) below your comments for revisions made in response.

-----

lines 37-38 BMI is subject to measurement error and has also been shown to underestimate weight status in values greater than 28

*This has been clarified further.

It is not clear to me that this statement is correct - does not the condition of obesity raise the risk of NCDs?

Line 43-44 - The research shows that this population is just as susceptible to the aforementioned chronic diseases as the general population [1,9-

*This has been clarified further.

Line 46- two of the most important contributors to weight status, nutrition status, and diet quality

More frequently the two most important contributors are level of active lifestyle and dietary patterns. - nutrition status and dietary quality are most often seen as subsets of the same contributing factor

*Nutrition status has been eliminated to focus on the 2 elements of importance to this study and add clarity.

Line 52-53- non-demographic related factors (non-homeostatic variables [17], does this include food environments? They have been proven to greatly impact food availability and food consumed.

*Agreed.  We are doing work with firefighters and the fire station food environment, which has a big impact on their habits. We have added “etc.” to show that these are not exhaustive lists.

Like the authors noted above about the criticism of the use of BMI, many critiques can be found of the Dietary Guidelines for Americans

Line 59-60 Dietary Guidelines for Americans which updates every 58 5 years and has been shown to be both valid and reliable amongst the US population for people two years and older [23].

*A comma has been added to show that the valid and reliable element is in relation to the HEI, not the DGA. We do acknowledge critiques are available for the DGA as well.

The rationale for using the two terms: nutrition status, and diet quality are not illuminated for the reader in the final two paragraphs of the introduction. If sufficiently distinct in this study, readers may benefit from understanding this distinction and application in this study.

*A statement has been added near the purpose to clarify the distinction and rationale for doing so.

This reviewer appreciates the complication but wonders about the methodology and the researchers not seeing or assessing examples of servings that were plated and the amount of food left unconsumed (or wasted).

*That is a potential next step, if we can find enough hands to do the work on location.  This study provides a best-case scenario of what could be eaten, theoretical start if you will.  Doing a plate waste study would add detail on what is truly consumed.

Line 113 this data - should this be these data. Are data not plural?

*This has been corrected.

Line 114-115 unclear - what need of cadets? And for normal cadet features does gender factor into weight and height and BMI?

*Some words and phrases have been added to improve clarity. The academy has a major physical training component, so the needs are the physical training (PT) performance nutrition needs of the cadets to fuel high level performance in and recovery from PT.

Line 155 - what is used to determine whether meeting average cadet needs for calories? For the food to not meet caloric levels on a daily basis seems surprising as the contrary would have been more likely. 2900 calories may be excessive for sedentary activity like attending courses for weeks. But it may be a good proxy for a more active lifestyle or an athletic population

*A statement about being provide details on the physical training regimen has been added to the methods for determination of calorie needs in combination with average cadet characteristics.

It may be helpful to note the importance of the quantity and the quality of the six nutrients that did not meet adequacy are integral for performance needs,; 179 these six nutrients were calories, carbohydrates, vitamins D and E, magnesium, and po- 180 tassium [27]

*To keep with your point earlier of not confusing the reader between nutrient adequacy and dietary quality, we would like to keep it to adequacy for this statement.  Dietary quality was emphasized earlier in the paragraph.  This statement addresses the secondary portion of the purpose.

Reviewer 4 Report

This study is interesting and well written. There are certain limitations of the study, but they are fully addressed in the manuscript.

I have only a couple of suggestions for a clearer presentation of the results:

a) Line 20: “The total HEI score was 54”. I suggest that you add an interpretation of this result to be clear what does it mean.

b) Regarding the statistical analysis, have you tested the normality of the data? How did you decide to use the parametric t-test?

Author Response

Thank you for your time in reviewing this manuscript and for your thoughtful comments. For (a), “/100” was added to provide context. For (b), we did ensure assumptions of parametric tests were met.  A phrase has been added to clarify.